# Minimizing isotropic and deviatoric membrane energy – An unifying formation mechanism of different cellular membrane nanovesicle types

**Veronika Kralj-Iglič**[1,2]*, **Gabriella Pocsfalvi**[2], **Luka Mesarec**[3], **Vid Šuštar**[4], **Henry Hägerstrand**[5,6], **Aleš Iglič**[2,3]

1 Faculty of Health Sciences, Laboratory of Clinical Biophysics, University of Ljubljana, Ljubljana, Slovenia, 2 Extracellular Vesicles and Mass Spetrometry Group, Institute of Biosciences and Bioresources, National Research Council of Italy, Napoli, Italy, 3 Faculty of Electrical Engineering, Laboratory of Physics, University of Ljubljana, Ljubljana, Slovenia, 4 Faculty of Medicine, Lymphocyte Cytoskeleton Group, University of Turku, Turku, Finland, 5 Faculty of Science and Engineering, Cell Biology, Åbo Akademi University, Åbo/Turku, Finland, 6 Novia University of Applied Sciences, Ekenäs, Finland

* veronika.kralj-iglic@fe.uni-lj.si

**Data Availability Statement:** All relevant data are within the manuscript.

**Funding:** EU Commission H2020 Ves4us 801338 (VKI, GP, AI) Slovenian Research Agency P3-0388,

## Abstract

Tiny membrane-enclosed cellular fragments that can mediate interactions between cells and organisms have recently become a subject of increasing attention. In this work the mechanism of formation of cell membrane nanovesicles (CNVs) was studied experimentally and theoretically. CNVs were isolated by centrifugation and washing of blood cells and observed by optical microscopy and scanning electron microscopy. The shape of the biological membrane in the budding process, as observed in phospholipid vesicles, in erythrocytes and in CNVs, was described by an unifying model. Taking the mean curvature $h$ and the curvature deviator $d$ of the membrane surface as the relevant parameters, the shape and the distribution of membrane constituents were determined theoretically by minimization of membrane free energy. Considering these results and previous results on vesiculation of red blood cells it was interpreted that the budding processes may lead to formation of different types of CNVs as regards the compartment (exo/endovesicles), shape (spherical/tubular/torocytic) and composition (enriched/depleted in particular kinds of molecules). It was concluded that the specificity of pinched off nanovesicles derives from the shape of the membrane constituents and not primarily from their chemical identity, which explains evidences on great heterogeneity of isolated extracellular vesicles with respect to composition.

## Author summary

One of the amazing properties of a biological membrane is the ability to undergo dramatic changes of its shape. It may exhibit very high curvature and thereby enclose nano-sized compartments that pinch off from the mother membrane and become freely moving cellular nanovesicles (CNVs). CNVs externalize the pieces of the cell and make them

J1-9162, L3-2621 (VKI) Slovenian Research
Agency P2-0232 (LM,AI)

**Competing interests:** The authors have declared
that no competing interests exist.

**Abbreviations:** CNVs, cellular nanovesicles; EVs,
extracellular vesicles; ESCRT, endosomal sorting
complexes required for transport.

available to other cells within the same organism or other organisms. Therefore they have
been acknowledged as mediators of communication between microorganisms, plants, ani-
mals and human. Furthernore, they dwell on the border between living and non-living
things. Recent findings report on heterogeneity of the size and composition of CNVs
found in isolates from different biological samples. As communication between cells is
involved in many physiological and patophysiological processes, it is of importance to
understand the mechanisms of CNVs formation and recognize the natural laws that
mainly govern them. We point to an unifying mechanism that explains stability of differ-
ently shaped and composed CNVs by taking into account that the biological membrane
tends to attain the minimum of its relevant energy. Conveniently, the procedure can be
described by a mathematical model which allows for transparent comparison between
experimentally induced shapes of membrane-enclosed vesicular structures and numerical
calculations.

## Introduction

Submicron sized membrane-enclosed cellular fragments that can mediate interactions
between cellular compartments, cells and organisms have recently rised high hopes for diag-
nostics and therapy of different diseases. Understanding mechanisms of their formation is of
utmost importance for effective use in science, medicine and technology [1–2]. In particular,
the discovery of universal mechanisms explaining the phenomena appears to be "crucial and
highly warranted" [3]. The analyses of the micro and nano particles isolated from different bio-
logical samples are informative, but do not essentially explain how the particles were created
[4]. It is therefore necessary to study the processes leading to the release of the vesicles from
the membrane.

  Mammalian erythrocytes and giant phospholipid vesicles (that lack internal structure) have
been used as model systems to study principles of membrane budding (the process in which
the membrane wrinkles to form a precursor of the vesicle) and vesiculation (the release of a
stable vesicle from the mother membrane). The features taking place in the membrane are
reflected in the cell/vesicle shape (on the mesoscopic range) which can be observed by different
microscopic techniques and compared to theoretically calculated shapes [5–7]. The mecha-
nisms conveniently studied in these simple systems can then be generalized to other types of
biological membranes as they all share common essential properties.

  Membrane vesiculation has been extensively studied experimentaly and theoretically for
the last 30 years [8,9]. The early studies described the membrane shape by minimization of its
elastic energy [5,10] while shape transformations were explained by the change of the differ-
ence between the two membrane layer areas [11–13]; if the outer layer expands relative to the
inner one, the membrane wrinkles outwards and vice versa. Correspondence between the
observed and the calculated shapes was reported in numerous studies that elaborated various
phase diagrams of possible shapes (see for example an extensive review of early work by U. Sei-
fert (1997) [14]). Also inhomogeneities of the membrane composition were taken into account
in theoretical descriptions and observed in experimental systems [15–20].

  The above early theoretical models considered mildly curved membrane surfaces which
could be well described by viewing the membrane as a thin laterally isotropic elastic shell. In
membranous nanostructures, however, the membrane is strongly curved and the curvature
radii approach the sizes of the membrane building units. Self-assembly of the molecules needs
to be considered [21]. It is relevant to theoretically describe such membrane as composed of

many constituents and apply the methods of statistical physics. This reveals a necessity of consideration of in-plane anisotropy and orientational ordering of membrane components and therefrom deriving deviatoric elasticity of the membrane [16].

Evidences from different assessment techniques (flow cytometry, electron microscopy, light scattering, immunolabeling) indicate the presence of submicron sized particles in different biological samples. Recently, these particles were nominated »extracellular vesicles« (EVs) [22] and the International Society for Extracellular Vesicles (ISEV) that was founded in 2012 has put forward a set of conditions (i.e. experimental methods used) which qualify material as EVs [23]. Furthermore, within this view, it is presently acknowledged that there are three types of EVs: apoptotic bodies, microvesicles and exosomes (see for example [24]); it is suggested that the apoptotic bodies are formed during the decay of the cell undergoing apoptosis, microvesicles are formed by exovesiculation of the plasma membrane while exosomes are formed in a complex process involving formation of an endosome into which nanovesicles are shed and released from upon fusion of the endosome with the plasma membrane. Although there is a distinction between these modelled processes, the particles in actual isolates may come from any of them as well as from some other processes (e.g. formation of viruses or fragmentation of cells due to mechanical, thermal and chemical stress during the processing). Virions are within the same range of sizes as EVs and the enveloped ones have a membrane, just like EVs [25]. Also understanding processes of formation of virions is important for revealing appropriate drug targets in virus-associated diseases [26].

A view on tiny membrane-bound vesicular structures can therefore be taken from the point of membrane. The scope of this work are cell membrane nanovesicles (CNVs): submicron sized membrane-enclosed cellular fragments that have been formed in a process in which membrane plays a key role. CNVs include microexovesicles, exosomes, enveloped viruses and cellular membrane endovesicles. The main distinction between the term »cellular nanovesicles« and the term »extracellular vesicles« is that »cellular nanovesicles« primarily refer to the isolates (particles formed by diverse processes which are enclosed by the membrane that determines their shape) while the term »extracellular vesicles« refers to the modeled *in vivo* mechanisms taking place in cells according to the presumed processes and are presumably harvested as such.

In this work we aim to relate an universal mechanism of the membrane budding and vesiculation to processes of formation of differently shaped and composed micro/nanovesicles. We hypothesize that formation of all types of CNVs is based on the same biophysical mechanism, i.e. minimization of the membrane free energy that includes isotropic and deviatoric elasticity. The process is reflected in the change of the membrane shape due to changes in the membrane environment and may eventually lead to the pinching off of the micro/nano vesicles from the mother membrane. The model is validated by comparison with experimentally observed vesiculation. The experimental results involve nanovesicles of different kinds: as regards the compartment into which the vesicles emerge (exovesicles, endovesicles) and as regards general shape (spherical, tubular and torocytic).

## Theoretical background

In membrane-enclosed entities filled with liquid without internal structure, the shape of the vesicle is largely determined by the properties of the membrane. Such property is the local curvature of the membrane. In a chosen point P of the membrane surface we can construct a vector **n** pointing in the direction perpendicular to the surface (Fig 1A). This vector is called the normal. If we cut the surface with a plane through the normal (Fig 1A) we get a curve. When the shape of the curve is fitted in the chosen point with a circle (as demonstrated in three

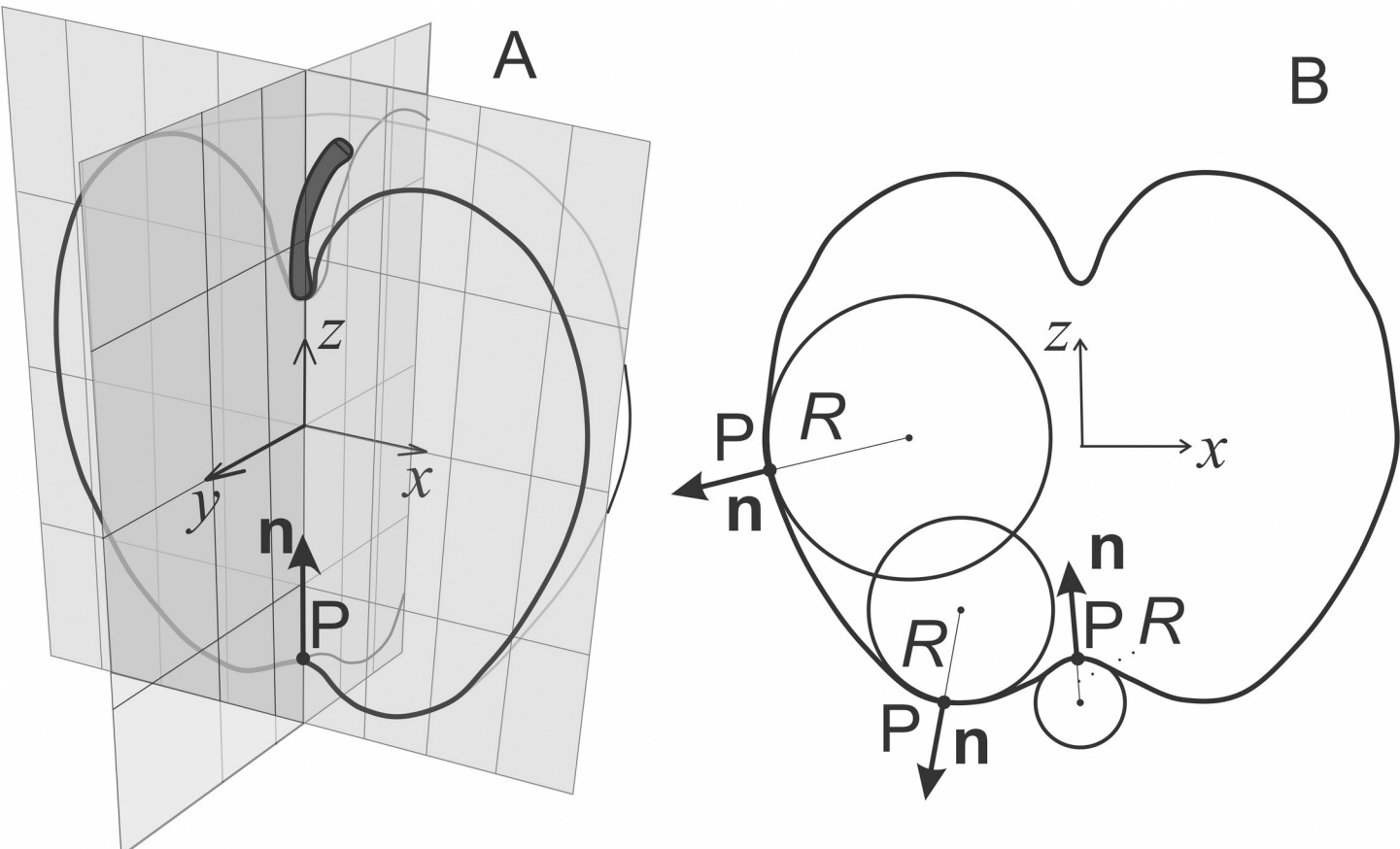

**Fig 1. A scheme of an area element.** A: A normal cut through the vector **n** that is perpendicular to the surface (the normal) at the point P (the normal). B: Front view of the normal cut with three chosen points on the cut (P), the respective normals (**n**) and curvature radii (R).

chosen points in Fig 1B), the curvature of the respective normal cut is given by the reciprocal value of the radius of this circle (R), $C = 1/R$.

Membrane curvature that fits a given membrane constituent is called the intrinsic curvature of the constituent. We propose that the mean curvature of the membrane surface

$$H = (C_1 + C_2)/2 \tag{1}$$

and the curvature deviator

$$D = (C_1 - C_2)/2 \tag{2}$$

and the respective intrinsic quantities of the membrane-building elements ($H_m$ and $D_m$) are key parameters in modeling of the mechanisms underlaying budding and vesiculation of the membrane and therefore also of the equilibrium shape and composition of the cellular micro/nano vesicles. Some characteristic local shapes of the membrane are depicted in Fig 2A–2D.

In accordance with the fluid crystal mosaic model [16], membrane is considered as composed of constituents (inclusions) subjected to the local curvature field created by surrounding constituents. In order to compose the membrane, the constituent attains the local membrane shape that usually differs from its intrinsic shape. The principal axes of the membrane and of the constituent are in general rotated by an in-plane angle ω, meaning that the constituents can attain different in-plane orientations in the membrane which correspond to different

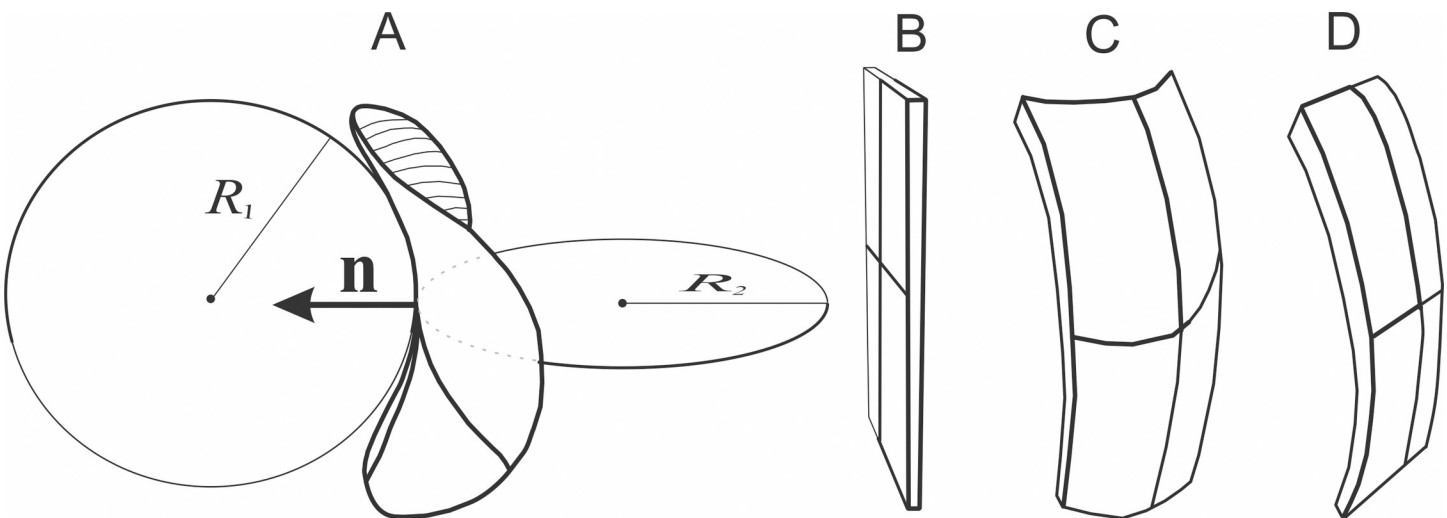

**Fig 2. An area element of the membrane with the normal (indicated by a vector n) and the principal directions.** The radii of the circles that fit the shape of the normal cuts in the principal directions at the chosen point are shown and the radii $R_1$ and $R_2$ determining the local principal curvatures $C_1 = 1/R_1$ and $C_2 = 1/R_2$, are indicated. Possible local shapes of a membrane favoring A: Saddle–like ($H \neq D \neq 0$), B: Flat ($H = D = 0$), C: Globular ($H > 0$, $D \simeq 0$ or $H < 0$, $D \simeq 0$) and D: Cylindrical ($H = \mp D \neq 0$) shape.

energies. The thermal motion oposes the complete orientational ordering in the direction with the lowest energy but the constituents will spend on the average more time in the orientation with lower energy. The single-constituent energy expresses a mismatch between the parameters of the membrane at the position of the constituent ($H$ and $D$) and the intrinsic parameters of the constituent ($H_m$ and $D_m$) including mutual in–plane orientation of their principal axes [16]. The free energy of the whole membrane is obtained by summing up (integration) the contributions of the constituents and using methods of statistical physics [16].

If the membrane is composed of different types of constituents with given $H_{i,m}$ and $D_{i,m}$, where index $i$ denotes the type of the constituent, the above assumptions result in the expression for the free energy $F$ of the membrane

$$F = -kT \int \sum_i m_i \ln \left( q_i^0 \, 2 \cosh \left( d_{i,\text{eff}} \right) \right) \mathrm{d}A + k_\text{B} T \int \sum_i m_i \ln \left( m_i / m \right) \mathrm{d}A \qquad (3)$$

$$q_i^0 = \exp \left( -\frac{\xi_i (H - H_{i,m})^2}{2k_\text{B}T} - \frac{(\xi_i + \xi_i^*) \left( D^2 + D_{i,m}^2 \right)}{4k_\text{B}T} \right) \qquad (4)$$

$$d_{i,\,\text{eff}} = (\xi_i + \xi_i^*) D D_{i,m} / k_\text{B} T \qquad (5)$$

where $\xi_i$ and $\xi_i^*$ are constants, $k_\text{B}$ is Boltzmann constant, $T$ is temperature, $m_i$ is local area number density of the $i$-th kind of constituents and $m$ is number of all constituents divided by the membrane area. Integration is performed over the membrane surface $A$ and summation is performed over all types of constituents (index $i$ runs over all types of constituents).

It is convenient to state the model equations in dimensionless form. Dimensionless quantities are introduced: $c_j = C_j R$, $j = 1,2$ are the principal curvatures normalized with respect to the radius of the sphere with the surface area $A$, $R = \sqrt{A/4\pi}$; $h = HR$ and $d = DR$ are the normalized mean curvature and curvature deviator of the membrane, respectively, and $h_{i,m} = H_{i,m}R$ and $d_{i,m} = D_{i,m}R$ are the intrinsic mean curvature and the intrinsic curvature deviator of the $i$-

th type of constituents, respectively. The area element is normalized with respect to the area $A$, $da = dA/4\pi R^2$, while the free energy is normalized with respect to the free energy of the sphere composed of chosen type of constituents, $8\pi m\xi_i$.

It is assumed that the system will attain the configuration (shape and composition–if multi-component) which gives minimal free energy at given constraints (such as the required membrane area, the required enclosed volume and the required proportion of constituents). The mathematical procedure of finding the minimum of the membrane free energy at chosen constraints yields the equilibrium shape of the cell/vesicle. In mathematical terms, the solution requires stating the variational problem and solving it by some appropriate method (e.g. ansatz [5], numeric solution of differential equation [16,27], surface evolver [28] or finite element method [29]). The result is the equilibrium shape of the membrane which can be compared with the experimentally observed shapes.

Here, minimization of the free energy (Eq (3)) was performed numerically as described in the Methods. Below we show results for unicomponent and bicomponent membrane whereas generalization to multicomponent membrane is straightforward.

## Results

### Shapes with minimal free energy

To outline the principle of (inward and outward) budding, sequences of shapes corresponding to a formation of one (inward and outward, respectively) spherical bud were calculated by minimization of the free energy (Eq (6)) (Fig 3; sequences a-d and f-i, respectively). The corresponding shapes observed in CNVs (imaged by electron microscopy) and in giant phospholipid vesicles (imaged by optical microscopy) are shown.

It can be seen in different systems that theoretically calculated shapes and experimentally observed ones agree well over up to 3 orders of magnitude (the order of the size of giant phospholipid vesicles is between 1 and 100 μm, in erythrocytes it is about 5 μm and in CNVs it is from 30 nm to 1 μm). Note remarkable similarity of the discocyte shape of the CNV found in the isolate from blood (Fig 3A) and erythrocyte (Fig 3E), which is 10 times bigger. These results indicate a prevailing importance of the membrane and its physical properties in determination of the shape of the particles. Furthermore, identifying similarity with calculated shapes enables recognition of entities in the sample that are enclosed by the membrane and have no internal structure–the properties that define cell membrane nanovesicles. Since particles in the isolates can be observed by some microscopic techniques (CNVs in Fig 3 were imaged by the scanning electron microscope), the agreement of the observed and calculated shapes confirms unambiguously that the entities in the sample are vesicles. Fig 3J shows the budding erythrocytes. Precursors of the vesicles are formed on the top of echinocyte spicules. The size and shape of these buds agrees with the size and shape of nanovesicles found in the isolates, pointing to a process of shape transformation that precedes formation of the vesicle. Although the shapes of the sequence f—i are somewhat different than the shapes of the cells and evaginations shown in Fig 3J, the biophysical origin (i.e. the important role of the membrane) of the formation of the bud in the two systems seems essentially the same.

The calculated shapes are characterized by the relative volume $v = \sqrt{V^2/36\pi A^3}$ (where $A$ is the area of the vesicle membrane and $V$ is the enclosed volume), the average mean curvature $<h> = \int h \, da / \int da$ and the average curvature deviator $<d> = \int d \, da / \int da$. In the ($v$, $<h>,<d>$) phase diagram of possible equilibrium shapes (Fig 4), classes of shapes can be distinguished within the boundaries defined by a distinct variational problem of extreme $<h>$ and $<d>$ at fixed volume and fixed area. These shapes are spheres, cylinders and tori and their possible combinations, and pertain to gray curves in Fig 4. The trajectories of the

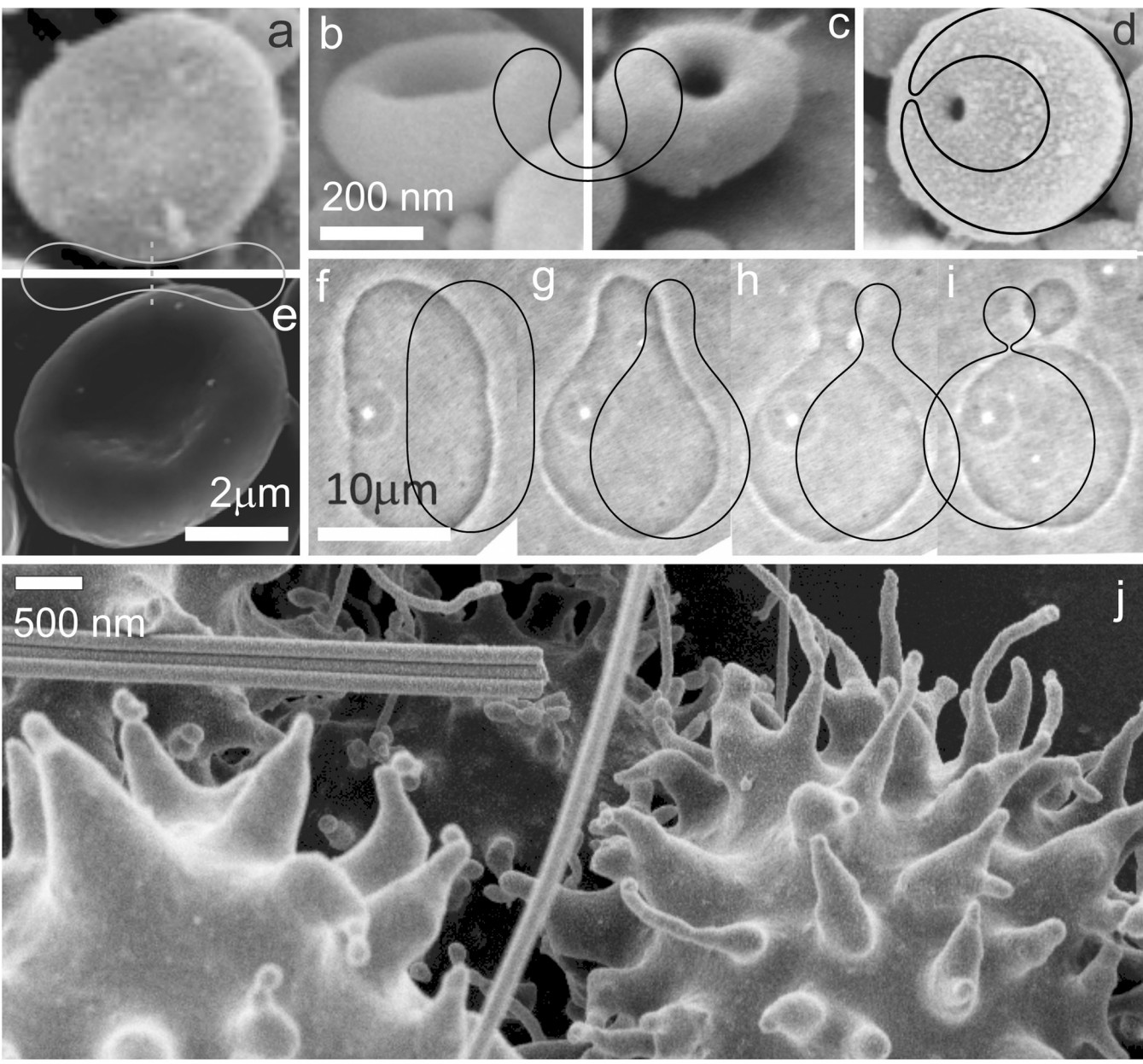

**Fig 3. Comparison between experimental and theoretical shapes.** a-d: The transformation of a discocyte into a stomatocyte as represented by the shapes observed in cellular nanovesicles, and the corresponding contours of the calculated shapes obtained by minimization of the membrane free energy. e: Discocyte shape of an erythrocyte complying with discocyte shape of cellular nanovesicle and with the calculated shape. f-i: The transformation of the outward bud in giant phospholipid vesicles and the corresponding result of the theoretical description obtained by minimization of the membrane free energy. The parameters of the calculated shapes are $h_m = d_m = 0$, (a,e): $v = 0.6$, $<h> = 1.040$, $<d> = 1.812$, (b,c): $v = 0.6$, $<h> = 0.650$, $<d> = 1.167$, (d): $v = 0.6$, $<h> = 0.435$, $<d> = 0.235$, (f): $v = 0.9$, $<h> = 1.050$, $<d> = 0.729$, (g): $v = 0.9$, $<h> = 1.105$, $<d> = 0.697$, (h): $v = 0.9$, $<h> = 1.155$, $<d> = 0.577$, (i): $v = 0.9$, $<h> = 1.240$, $<d> = 0.163$.

processes represented by the shape sequences can be visualized in the ($v$, $<h>$, $<d>$) phase diagram (Fig 4). Two perspectives of the phase diagram are shown for better visualization of the class boundaries and representations of possible processes (e.g. due to change of temperature or composition of the membrane that would affect $<h>$ and/or $<d>$). The outward budding (marked by blue) starts from a cylinder with hemispherical caps which with increasing $<h>$ and decreasing $<d>$ transforms into a pear shape, develops a neck and finally

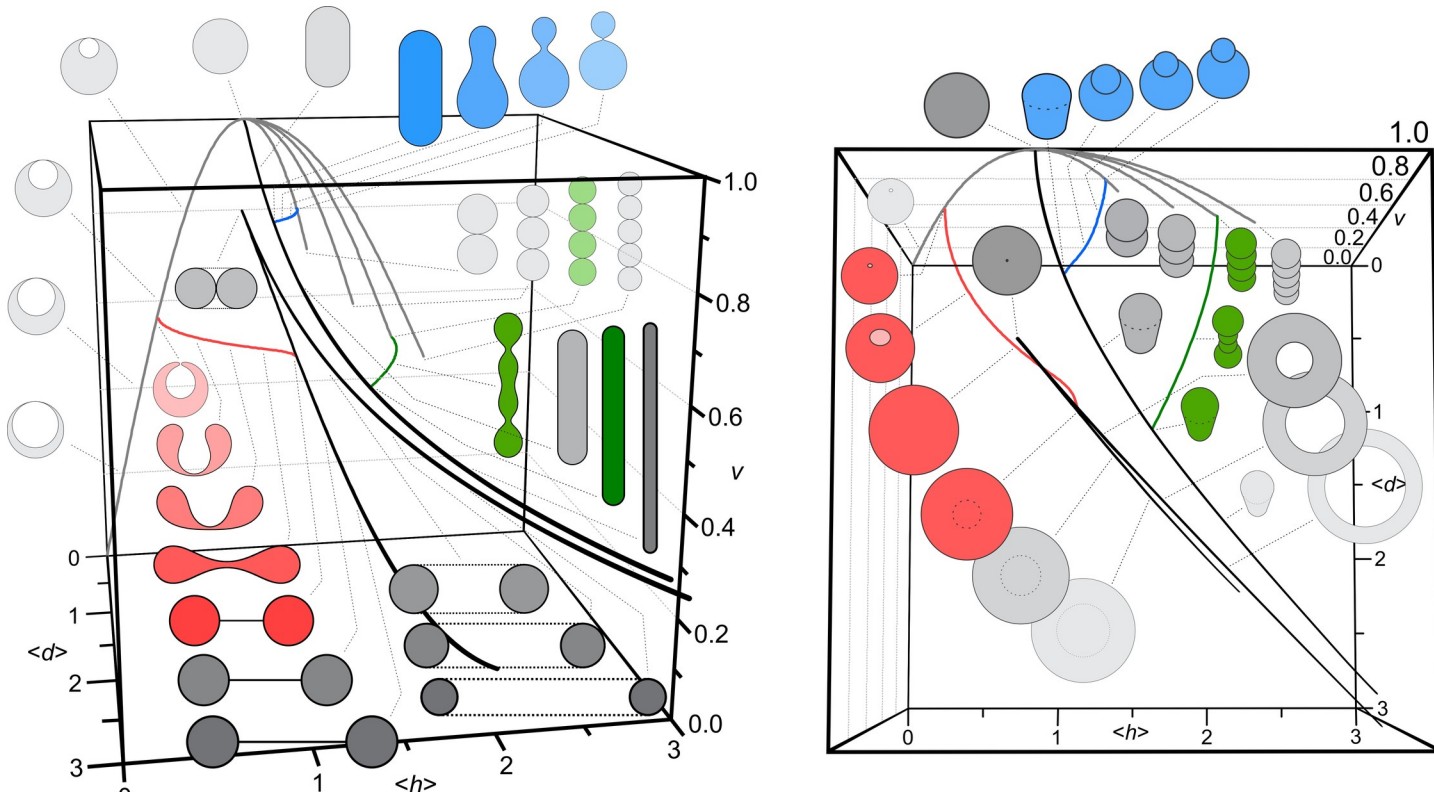

**Fig 4. Two aspects (side view (left) and top view (right)) of a three dimensional.**

yields two quasispherical vesicles connected by an infinitesimally thin neck (blue lines in Figs 4 and 3F–3I). A limiting point in the phase diagram is reached for a shape composed of the mother sphere and a spherical bud, where $< h >$ attains a local maximum and $< d >$ attains its minimal value 0 (Fig 4). The red lines in the phase diagram represent the process of formation of a spherical endovesilcle. Starting from a torocyte (torus with a flat membrane within the dimple) the shape transforms with diminishing $< h >$ and $< d >$ to reach an invaginated globule (stomatocyte) (red lines and shapes in Figs 4 and 3A–3D). Further, a neck forms and becomes thinner to finally yield a spherical endo-bud connected with the mother membrane with an infinitesimal neck (Fig 4).

**($v$, $< h >$, $< d >$) phase diagram.** Shapes composed of spheres, cylinders and tori (shaded gray) and sequences of shapes showing transformations are depicted.The transformation from a shape composed of a cylinder with hemispherical caps to a vesicle with evaginated sphere is represented by blue color, transformation of a cylinder with semispherical caps to four spherical beads is represented by green color, while transformation from torocyte shape to invaginated sphere is represented by red color. The shapes were calculated by taking $h_m = d_m = 0$, for which the solutions are invariant to scale.

Most of the shapes (aside from those which are composed of spheres) extend into the $< d >$ dimension of the phase diagram (Fig 4). It can be seen from the three dimensional ($v$, $< h >$, $< d >$) phase diagram that these parameters express the shape change and are therefore important in the processes of budding and vesiculation. However, to understand the direction in which the processes will develop, the solution of the variational problem gives also the values

of the free energy. It is assumed that the transformation will proceed in the direction of free energy decrease (provided that the constraints are accounted for).

## Curvature—induced lateral sorting of membrane constituents

In cells, membranes are composed of different molecules (characterized by different $h_{i,m}$ and $d_{i,m}$) which can also interact between themselves. As long as the membrane thickness is small comparing to the curvature radii of the cell surface, the inhomogeneities in composition are unlikely to be connected to the curvature of the cell surface. In the membrane budding, however, the membrane surface curvature increases considerably. Regions of energetically favorable curvature for membrane constituents with particular $h_m$ and $d_m$ are formed. As the membrane is more or less fluid in the lateral dimension, redistribution of different constituents may take place as regards the position as well as orientation of the anisotropic inclusions (if $d_m \neq 0$). We call this effect the »curvature sorting of membrane constituents«. For isotropic inclusions, the curvature sorting decerases the free energy (stabilizes the shape) by providing an additional degree of freedom to the system (i.e. lateral distribution of constituets). For anisotropic constituents, there are two degrees of freedom which can stabilize the shape: lateral distribution and the orientational ordering of constituents. The constituents lower the free energy (form and stabilize the shape) by moving to regions with curvature closer to the intrinsic ones, by orienting in the plane in such way to approach the orientation of the membrane (both to an extent consistent with entropic effects) and in turn, by creating areas with favorable curvature and average orientational order of constituents.

Fig 5 shows examples of self-consistent solutions of the variational problem yielding the equilibrium shape and the corresponding composition of a bi-component membrane. If the membrane is composed of two types of constituents (the majority favoring mild curvature and a smaller portion favoring strong curvature), separation of the constituents is indicated with

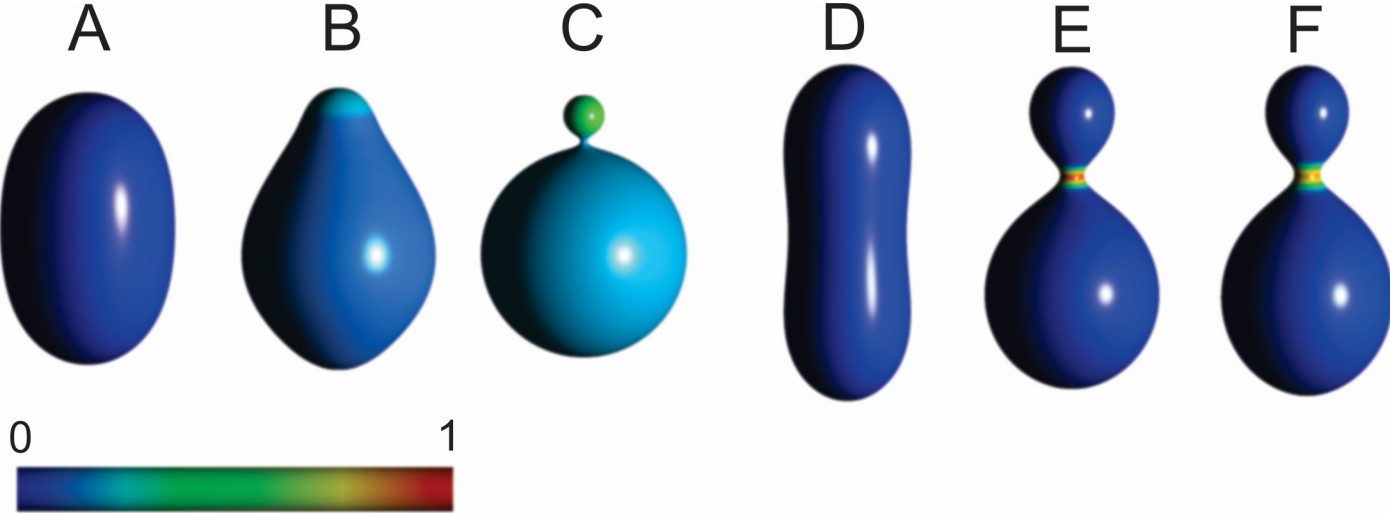

**Fig 5. Examples of calculated shapes and respective distributions of constituents in bi-component systems.** In all shapes: $k_1 = 30\ kT$, $k_2 = 240\ k_B\ T$, $R = 250$ nm. A-C: The membrane is composed of constituents type 1 that favor isotropic (Fig 2C) mildly curved membrane (dark blue) and constituents type 2 that favor isotropic (Fig 2C) strongly curved membrane (red), $h_{1,m} = 1$, $h_{2,m} = 16$, $d_{1,m} = d_{2,m} = 0$. Relative proportions of type$_2$/type$_1$ constituents are 1 (A), 0.125 (B) and 0.2 (C). D-F: The membrane is composed of type 1 constituents that favor isotropic (Fig 2C) mildly curved membrane and type 2 constituents that favor saddle shaped (Fig 2A) membrane, $v = 0.80$, $h_{1,m} = 2$, $h_{2,m} = 0$, $d_{1,m} = 0$, $d_{2,m} = 8$. Relative proportions of total amount of constituents (type 2/type 1) are 0.02 (D), 0.035 (E) and 0.045 (F). Colors representing local relative proportions of constituents are depicted in the scale. The parameters of the calculated shapes are (A): $v = 0.95$, $<h> = 1.0221$, $<d> = 0.55052$, (B): $v = 0.95$, $<h> = 1.0316$, $<d> = 0.64436$, (C): $v = 0.95$, $<h> = 1.1608$, $<d> = 0.17617$, (D) $v = 0.80$, $<h> = 1.1264$, $<d> = 1.1012$, (E): $v = 0.80$, $<h> = 1.2811$, $<d> = 0.67685$, F: $v = 0.80$, $<h> = 1.2760$, $<d> = 0.73304$.

changing proportion of the two species, according to the respective preferred curvature. With increasing portion of the constituents that favor strong curvature, budding of a smaller sphere is promoted (Fig 5A–5C) with notable redistribution of the components. If the membrane is composed of majority of isotropic constituents and a small proportion of constituents that favor saddle shape (Figs 2A and 5D–5F), a thin neck is formed upon increasing the proportion of constituents that favor the saddle shape and these constituents accumulate in it (Fig 5E and 5F).

## Discussion

### Liquid crystal mosaic provides explanation for various types of cellular micro/nano vesicles

In Fig 6 we attempt to interpret processes leading to different types of cell-derived vesicles by the same unifying mechanism as well as to present the images of the vesicles themselves. Fig 6A and 6B show images of apoptotic bodies and a decaying apoptotic cell. It can be seen in Fig 6B that the entire cell undergoes a decaying process thereby forming particles that likely include remnants of the internal structures (Fig 6A). Even if the shed particles organize by surrounding themselves with the membrane, the shape of the apoptotic bodies does not attain a regular form corresponding to the minimum of the membrane free energy. Namely, in such particles, the internal structure plays an important role in determination of the shape. This is out of scope of the theory considered in this work.

Spontaneous membrane budding and vesiculation in the physiological *ex vivo* conditions is a relatively rare event and for study purposes, methods have been developed to stimulate it with externaly added substances [8,30,31]. It was observed that at sublytic concentrations of various amphiphilic molecules added to the erythrocyte suspension, the erythrocytes undergo budding and vesiculation. Different pathways of the transformation were distinguished (Fig 6, sequences C-D-E-F; C-D-E-G, C-H-I-K; C-H-J-L); the particular pathway that takes place depends on the species and concentration of the added amphiphilic molecules [8,30,31]. In the pathway represented by Fig 6 C-H-I-K (in which erythrocytes were treated by dodecylzwittergent) discocytes transform into echinocytes and then into spheroechynocytes while at the top of the spheroechynocyte spicules, outward budding of the membrane takes place with subsequent release of exovesicles that conserve the globular size and shape of the buds [31]. In the pathway represented by Fig 6 C-H-J-L (in which erythrocytes were treated by dodecylmaltoside) the buds and the vesicles attain tubular shape. In both the above pathways, the average mean curvature $<h>$ increases (exhibiting echinocytosis and exo-budding). However, at a certain point (where the local curvature reaches higher values) the two pathways become distinct in the approach to the respective limiting shapes; in the one leading to the spherical submicron exovesicles, $<h>$ continues to increase and $<d>$ decreases while in the one leading to the tubular exovesicles, both $<h>$ and $<d>$ increase. The bilayer couple principle [11–14] expressed by the change of the area difference (which is formally equivalent to $<h>$) can explain this shape transformation only up to mildly curved buds and cannot distinguish between spherical and tubular budding and vesiculation. The shape composed of a cylinder with hemispherical caps and a shape composed of a series of quasispherical beads connected by infinitesimal necks may have equal relative volume $v$ and the average mean curvature $<h>$, but differ considerably in $<d>$ (Fig 4). The cylinder having the same area and enclosing the same volume as the series of spherical beads connected by infinitesimal necks has smaller radius than the series of spherical beads and its contribution of the bending elasticity to the free energy is larger. However, in the shape with cylinder (but not in the shape composed of spheres), there is also a contribution of the deviatoric term due to anisotropic curvature of the surface and consequent orientational ordering of the consituents. This contribution to the

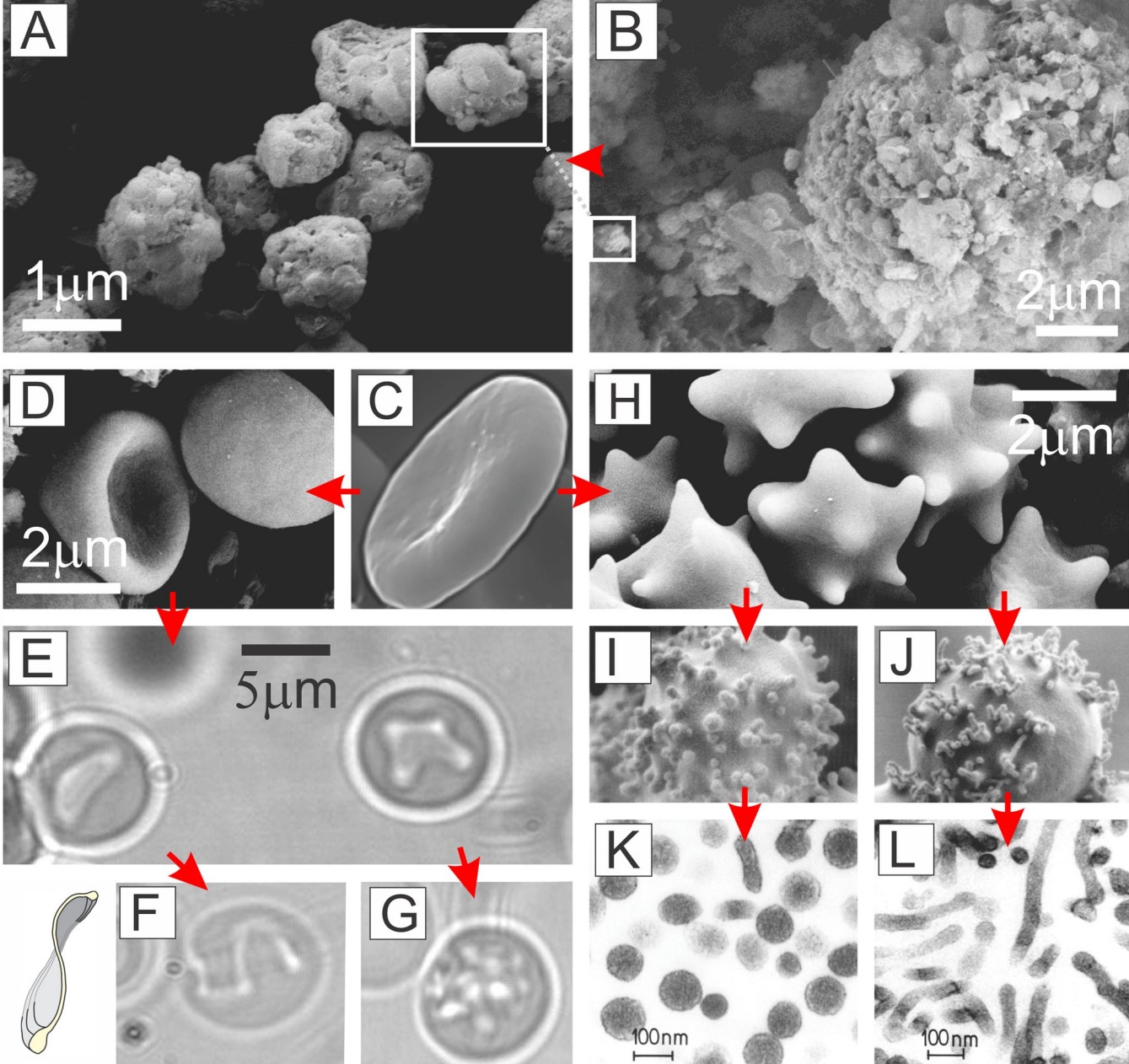

**Fig 6. Externally induced cell shape transformations and respective vesicles.** A: Apoptotic bodies and B: Apoptosis in Madin Darby Canine Kidney (MDCK) cell line, induced by staurosporine; C: A normal discocytic shape of a human erythrocyte at physiological *ex-vivo* conditions, D: Stomatocytes, E: Formation of folds in stomatocytes, F: Formation of a torocytic internal bud, G: Formation of multiple internal buds by externally added lidocaine (E-G), H: Echinocytes, I: Spheroechinocyte with glubular buds induced by dodecylzwittergent, J: Spheroechinocyte with tubular buds induced by dodecylmaltoside, K respective isolated spherical nanovesicles, L: Respective isolated tubular nanovesicles.

energy is always negative. If the cylinder is thin enough, the negative term may prevail and render that shape energetically more favourable than the shape composed of spherical beads [32]. The estimated dimensions of the shapes with stable phospholipid tubular protrusions are below 100 nm [32].

In another process (where erythrocytes were treated by ethyleneglycol C12E8), discocytes with decreasing $<h>$ transformed into stomatocytes and underwent an inward budding

leading to peculiar flat torocyte-shaped endovesicles with a toroidal bulby rim and a thin disc-like flat region filled with solution between the two membranes [33]. In such process, the stomatocyte dip becomes flattened and twisted upon action of externally added molecules (Fig 6E) with concomitant decrease of $< h >$ and increase of $< d >$. Addition of chlorpromazine to the suspension of erythrocytes, in turn, caused stomatocytosis with decreasing $< h >$, which was followed by formation of many small endovesicles of peculiar shapes [8,33]. The endpoint of small spherical endovesicles is subjected to decreasing $< h >$ and $< d >$. The endo-buds were found close to the surface of the cell indicating a possibility that pinching off of the vesicles has not taken place [8]. Fig 6G shows development of many endo-buds folowing stomatocytosis.

Deviatoric elasticty explains existence of spontaneously stable spherical and tubular buds and exovesicles of the erythrocyte membrane (Fig 6K and 6L) and stable tubular phospholipid protrusions [32], but also of other structures with high anisotropy in membrane curvature (e.g. thin necks connecting the mother membrane and the bud [34] and nano-sized hexagonal and inverted hexagonal phases [35]). Recent experiments provided arguments in favor of the importance of protein orientation within the membrane on the CNV formation [36].

As membrane vesiculates and ireversibly looses a considerable part of its surface, the process ends with hemolysis and such transformation (also considered as an induced apoptosis in erythrocytes) is called eryptosis [37]. Since in erythrocytes, vesiculation of the plasma membrane can be clearly observed, a question can be posed wheter vesiculation of the erythrocyte plasma membrane is actually an essential process which takes part (also) in apoptosis and not a diverse process. Regardless of this classification, the induced vesiculation of erythrocytes is the scope of our work since it corresponds to the predicted behavior based on the minimization of the membrane free energy.

## Direct interactions between membrane constituents as a base for gross changes in membrane budding and vesiculation

Experiments showed that echinocytic transformation was prerequisite for the release of exovesicles, however, exovesiculation did not always follow echinocytosis [8]. The addition of $C_{12}$-alkyltrimethylammonium bromide has induced echinocytosis and exovesiculation which were followed by stomatocytosis [8]. These observations indicate that the preferrence for positive/negative curvature may change essentially during the process. It can also be observed (Fig 6) that the mean curvature of the exo and endovesicles is notably larger than the curvature of the membrane within preceeding echinocytes or stomatocytes, indicating dramatic changes in organization of the membrane at a certain point. Generalizing these results, it could be concluded that mild curvature change in the process of echinocytosis or stomatocytosis are followed by gross changes in organization of the membrane with either exo or endo–nanovesiculation–or both at the same time, depending on the local distribution of constituents.

We suggest that the mechanism causing the gross changes in the organization of the membrane (schematically shown in Fig 7) is based on direct interactions between the molecules that constitute the membrane. Curvature sorting of membrane constituents in the early phase enables the in-plane concentration of a certain type of constituents to reach a critical value where the direct interactions or chemical reactions between constituents take place. In Fig 7, a given type of membrane proteins is characterized by a blue part within the membrane, indicating a preference of these molecules for invaginated regions. Inward wrinkling of the membrane promotes accumulation of this type of molecules within the invagination. When their concentration reaches a certain threshold, membrane rafts or complexes are formed from

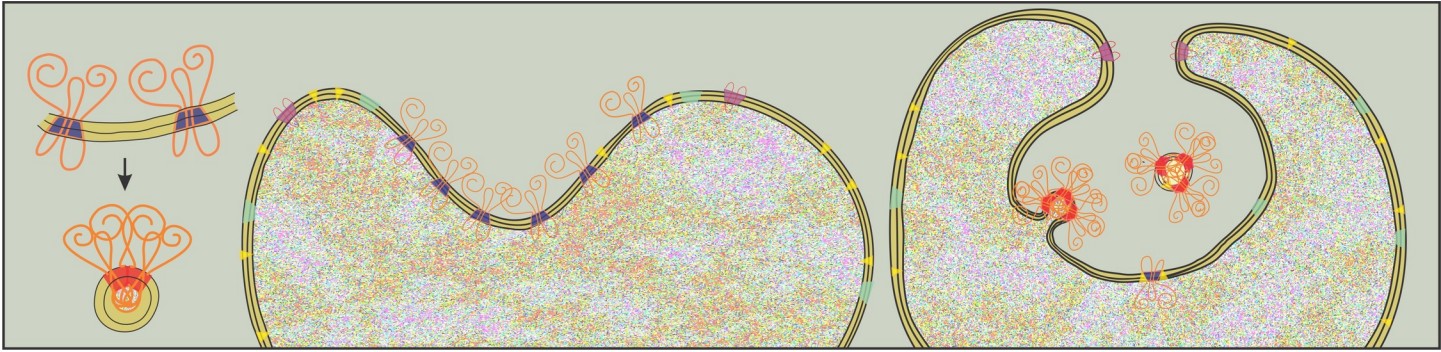

**Fig 7. Schematic change of intrinsic parameters $h_m$ and $d_m$ due to direct interactions between the molecules constituting the membrane.** Constituents with negative $h_m$ (blue part in the membrane) preferentially accumulate in the inward fold of the membrane. At high concentration of consituents with negative $h_m$ direct interactions become important and complexes are formed. The depicted complexes are characterized with strongly positive $h_m$ and the constituents form outward buds that may pinch off from the membrane.

these molecules, with the part within the membrane now prefering strong positive curvature. In other words, the identity of the constituents is re-defined by the change in their $h_m$ and $d_m$ (in Fig 7 this is indicated by the red color in the membrane part) and the pathway to curving the membrane is changed to nanoexovesiculation. Newly formed constituents with positive intrinsic mean curvature induce budding of the membrane outwards (to the cell exterior or into the invagination) while those with negative mean curvature induce budding of the membrane into the cell lumen. The suggested mechanism provides a possibility for the cell to shed the material out of the cell as well as a possibility to internalize the solution of the cell exterior or of the endosomal compartment.

Experimental evidences support the above reasoning; for example, it was observed that GM1-BODIPY membrane proteins accumulate in the invagination prior to membrane vesiculation [38]. Similar arguments were given related to the budding of the virions from the membrane; it was suggested that membrane protein dimerization was crucial for formation of the virion membrane [39]. Re-formation of membrane constituents within the budding and vesiculation process includes conformational changes of the molecules, changes of the conditions in the adjacent solution [40] and ATP-dependent mechanisms recently recognized as active forces [41,42]. Co-operative interactions between clathrin and endocytic proteins were suggested to be involved in formation of clathrin-coated vesicles [43].

### Exosome and virion formation

Mammalian erythrocytes attain their mature form devoid of the nucleus in a process of development from stem cells. Towards the end of the maturation the molecules that are involved in the hemoglobin production (e.g. transferrin which is an integral membrane protein) are discarded from the cell. It was observed that following invagination of the membrane, transferrin was shed from the folded membrane in the form of nano-sized globular vesicles [44–47]. These experiments brought an important contribution to recognition and definition of the third acknowledged mechanism of EV formation: release of so-called »exosomes« into the internal folds/cell compartments. Within the above theoretical and experimental reasoning, the formation of the exosomes in reticulocytes as recorded in evidence with transferrin release [44,45,47] could correspond to the process where the initial inward folding of the plasma membrane took place due to accumulation of membrane constituents with negative intrinsic mean curvature. When a certain threshold of the concentration of the transferrin-based inclusions was reached, gross reorganization of the components and formation of new complexes

took place thereby inducing exovesiculation into the compartment created by the invagination. In this view, the fold with the negative mean curvature served mainly to accumulate the molecules with negative intrinsic mean curvature in order to enable close approach between them and promote the formation of complexes. These complexes are then considered as constituents with their own characteristic $h_m$ and $d_m$ that are likely to be different than the intrinsic curvatures of the elements they were composed of (Fig 7). The self-assembly that is formed within the invagination is subjected to minimization of the membrane free energy. If the sign of the intrinsic mean curvature of the newly formed complexes ($h_m$) is positive, the outward budding of the membrane within the invaginated fold leads to the presence of vesicles in the invagination [46,47]. However, if the newly formed inclusions have negative $h_m$, inward budding (into the cell lumen) of the membrane within the invaginated fold can take place (as seen in Fig 1 of the reference [46]). Budding of the membrane of the internal compartment (endosome) inward and outward was recorded also in white blood cells [48].

## Curvature-sorting and composition of EVs

Biochemical functions of lipid rafts are mainly controlled by lipid-protein, lipid-lipid and protein-protein interactions as well complex interplays between the membrane and actin cytoskeletal meshwork and extracellular matrix. These interactions have transient and dynamic nature that makes their experimental studies very challenging. Molecular studies targeting the structural characterization of biomembranes and the interactions between membrane components are foremost important to understand membrane and vesicular genesis, functions and regulation.

Besides classical methods used in studies of molecular interactions (e.g. nuclear magnetic resonance, isothermal titration calorimetry, microscale thermophoresis, surface plasmon resonance and X-ray crystallography), new tools (i.e. native mass spectrometry, protein/lipid arrays, cryogenic-electron microscopy, Förster resonance energy transfer, attenuated total reflection Fourier transform spectroscopy; and methods that rely on labeling, chemical modification/crosslinking, e.g., lipid pull-down and lipid-overlay assay) and in silico computational methods have been developed and used to characterize non-covalent lipid-protein interactions [49,50]. Posttranslational modification of proteins by lipids is catalysed by specific lipid transferases. There are five lipid classes (fatty acids, isoprenoids, sterols, phospholipids, and glycosylphosphatidyl inositol anchors) that can modify covalently the proteins and thus confer different physiochemical properties to the proteins. Protein modification by lipids is important signal for the translocation and the reversible association of proteins to membranes as well as important mechanism for the regulation of diverse arsenal of protein functions. More than thousand proteins were reported to have lipid posttranslational modifications amongst which N-myristylation [51] and palmitoylation [52,53] have been suggested to regulate the membrane organization and to have a role in the biogenesis of EVs. MamProMD and GPS-lipid are databases and tools for *in silico* analysis of protein lipidation.

Existence of heterogeneous compartments in the plasma membrane was shown by fluorescence live staining using metabolic labeling of membrane proteins with organic fluorophores and click chemistry in combination with super-resolution fluorescence imaging by direct stochastic optical reconstruction microscopy [54]. More recently, the latter technique with photon switchable lipid dye has also been used for the characterization and visualization of uptake of EVs [55]. It was suggested [56] that rafts cover about the half of the plasma membrane surface in general and that the apical membrane of epithelial cells can be considered like a unique large raft. Membrane rafts can be isolated from the rest of the biomembrane based on their relatively low solubility in non-ionic detergents. Liquid chromatography and tandem mass

spectrometry-based methods have been found powerful in the analysis of detergent insoluble lipid raft preparations for the identification of post-translationally modified proteins (palmitoylated, myristoylated) present in the rafts. Highly expressed proteins in lipid raft isolates were found to be signaling proteins, like G protein-coupled receptors, protein kinases (e.g. Src family kinases) and phosphatases, glycosylphosphatidyl- inositol- linked proteins, caveolins, flotillins and transporters.

Interestingly, these proteins have also been identified/associated with EV preparations. It was shown that during physiological processes and progression of various diseases vesiculation of red blood cells and platelets took place at distinct membrane domains [57]. Dubois et al. (2015) [58] identified 350 proteins including tetraspanins, endosomal sorting complexes required for transport (ESCRT) proteins, and Ras-related proteins in lipid rafts of prostate epithelial cell-derived exosomes. Furthermore, a correspondence in protein/lipid composition was found between lipid rafts in erythrocytes and the extracellular vesicles isolated from the erythrocyte suspension [38].

Rafts can be considered as platforms for sorting of membrane constituents. Due to specific composition of rafts, also their local curvature may be different than in other membrane regions. Indeed, such formations also called caveolae were observed in experiments [59] indicating that curvature underlays a sorting mechanism connecting the shape and the composition of biological membranes. Recently, it was observed that endocytosis in diverse cell types was connected to formation of membrane domains [53]. This process was promoted by membrane proteins in these domains; in particular by their palmitoylation [53]. In the view of the suggested mechanism of exosome formation we think that the particular local composition of molecules that compose the membrane are essential in formation of the complexes that act as membrane constituents by a defined $h_m$ and $d_m$. Proteins and lipids and their associated complexes are important constituents and should be assessed to complement the characterization of the biomembrane. In this respect, characterization of the lipids is important [60]. Recent reports show the power of quantitative tandem mass spectrometry-based method for the untargeted analysis of lipid content of EV isolates. Using this method Chen et al. (2019) [61] showed that 53 (out of 422) independent lipid species are significantly different between the three buoyancy EV fractions purified by gradient ultracentrifugation from human serum.

## Mechanisms of CNV formation

On terms of the above, two processes leading to release of particles from cells can be distinguished. The first one follows cell decay in which three-dimensional structural elements surrounded by the membrane are organized into roughly-shaped globules (Fig 6A and 6B). The shape (and composition) of these particles is largely determined by the internal structure of the three-dimensional cytoskeleton remnants. The second process consists of budding and vesiculation of the membrane which is governed by minimization of membrane free energy (represented in Fig 6C–6L). The composition of vesicles shed from the membrane is determined by the local composition of the budding membrane and the enclosed part of the cell interior, where the key property is the mismatch between the local membrane curvature and the intrinsic curvature of the constituents. As there is a huge number of membrane constituents yet discovered, wide variety of the composition of enclosed adjacent cell interior [62] and different conditions/processing of samples applied [63–66], the composition and the size of EVs were found to be largely heterogeneous. Our results indicate that the intrinsic curvatures of the constituents are key factors in formation of EVs, therefore the mechanism would be non-specific primarily with respect to the chemical composition of molecules, but specific as regards their shape within the types shown in Fig 2.

We think that as regards physical properties, there is no essential difference between the so-called »microvesiculation of the plasma membrane« and formation of »exosomes« in multivesicular bodies. Both processes can be described by the same mechanism of an increase or a decrease of $< h >$ while the difference between them is due to different values of the $h_m$ and $d_m$ parameters of the underlying membrane which determine the size, shape and composition of buds and EVs. Buds with large curvature leading to small EVs can be shed from the evaginations and invaginations; however, formation of larger buds and EVs into the invaginations is limited due to lack of space. Once released from the cell, it is however in principle not possible to say whether a nano-sized vesicle has originated from the plasma membrane or an internal membrane. Furthermore, microvesicles and exosomes may originate from the same plasma membrane and are facing the interior and exterior of the same cell in the same orientation as regards the inner and the outer membrane layer.

It was suggested [67] that enveloped virions may be considered as a type of EVs since there are many essential similarities between virions and EVs: RNA viruses have a similar size as exosomes, follow ESCRT pathways [68] and have similar physicochemical properties and evolutionary roots; enveloped viruses could have been formed by encapsulating nucleic acids and incorporating specific membrane molecules while EVs could be viruses that have unsufficient machinery for replication. According to our mechanism, enveloped viruses can be considered cellular micro/nano vesicles with particular genetic cargo but subjected to the same unifying formation mechanism.

External vesiculation of the membrane allows for the use of the scanning electron microscope, and the buds and vesicles can be visualized also by the shape of the surface. The information of the surface topography cannot reveal the features inside the cell, so analysis of the endovesicle shape is limited to other techniques. From transmission electron images alone it cannot be concluded whether the structure within the cell contour is a large endovesicle or an invagination with contact to the outer solution. A simulation of different cuts through a cell with the internal fold (Fig 8) shows that a cut through the shape with internal fold may give an image of closed internal compartment although the internal fold may be connected with the outer solution. Also the internal compartments (e.g. endosomes) might not completely detach from the membrane but remain connected to it by membranous nanotubes. This seems possible and probable since it would not require energy-costly pinching off and fusion of the endosome with the membrane to enable the exosome release from the cell. Yet, the imaging techniques that require slicing of material could miss these thin structures. Similar features have been observed in giant phospholipid vesicles where dilatations of the thin membranous tubes may appear as vesicles [69]. As our knowledge on membranous nanostructures within the cell interior is yet rudimentary, continuation of the resarch of membrane-based mechanisms is necessary. Observation of membrane and vesicle shapes may contribute a significant part to understanding the formation of CNVs.

## Conclusions

Based on about 30 years of theoretical and experimental studies we propose a description of nano- vesiculation of membranes by the liquid crystal mosaic model of the biological membrane. Within this model the membrane is viewed upon as composed of constituents with intrinsic shape given by the mean curvature $h_m$ and the curvature deviator $d_m$. The model is supported by a mathematical formalism deriving from a single-constituent energy and applying methods of statistical physics. In the model, the constituents undergo lateral and orientational ordering within the membrane and compose the shape of minimal free energy at given constraints. Ample experimental evidence that has been gathered recently indicates that a subjacent pool of membraneous nanostructures (nanotubes, nonlamellar phases such as hexagonal, inverted hexagonal and cubic,

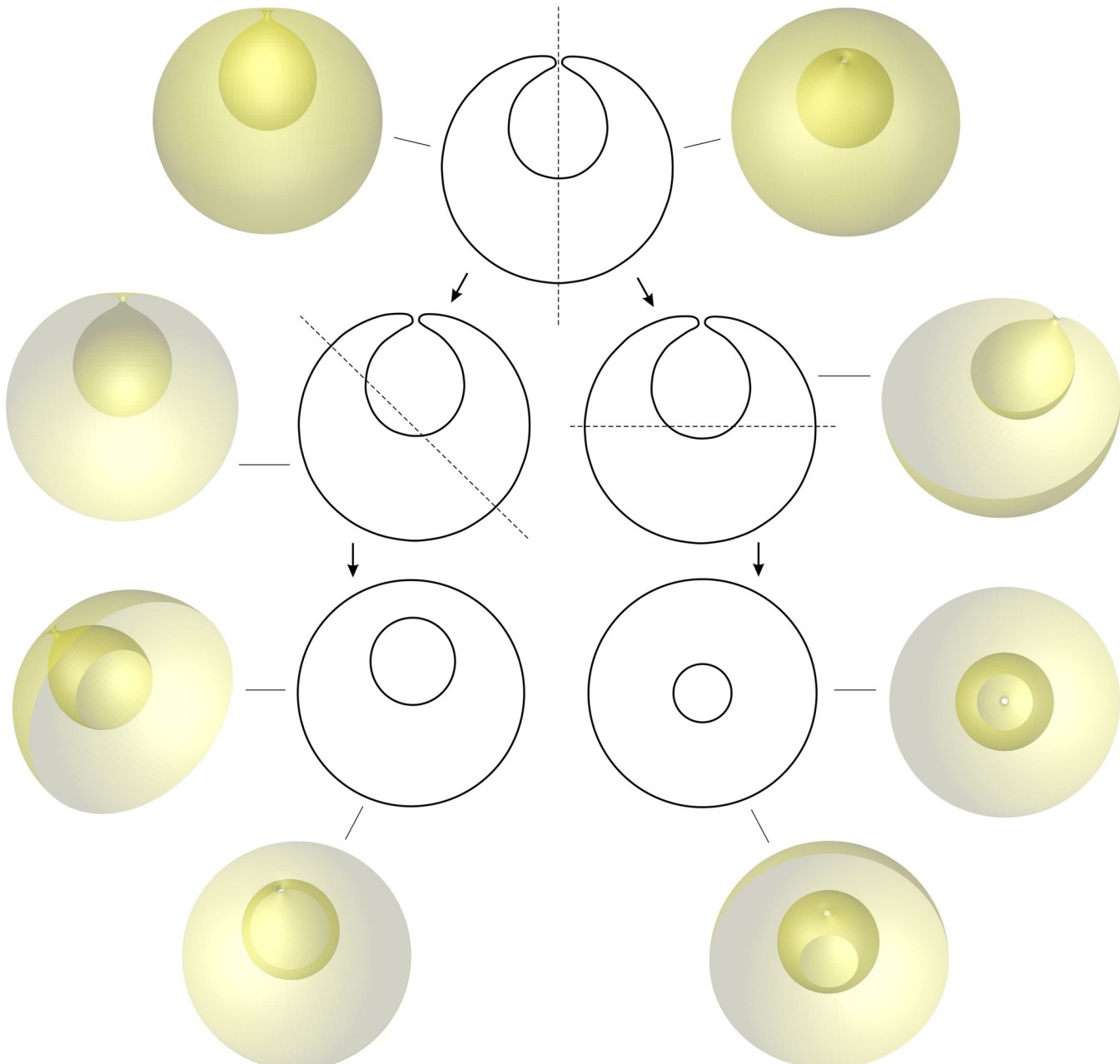

**Fig 8. Different cross sections of the hypothetical cell with an internal fold exhibiting the connection of the invagination (upper part) and hiding the connection (lower part).**

and toroidal vesicles) is a common feature in living systems and can no longer be overlooked in considering physiological processes taking place in cells and related systems.

Further we propose that formation of nano-sized exo and endo vesicles (including microvesicles, exosomes and viruses) is driven by essentially the same mechanism based on minimization of the free energy where direct interactions and formation of membrane constituents with specific principal intrinsic curvatures ($h_m$ and $d_m$) play important role. This biophysical

mechanism alows for the explanation of the expected large heterogenity of different molecules found in isolates from cell culture media and body fluids [70].

Correspondence between theoretically obtained shapes and the shapes of CNVs observed by the microscope evidences that the particles under the investigation are membrane–enclosed and have no internal structure. The curvature sorting and lateral and orientational ordering of membrane constituents which are based on rigorous physical methods provide simple and plausible explanations of some essential common features in membranous systems and should be taken into account in description of these important systems.

## Materials and methods

### Experimental

**Imaging of apoptosis and apoptotic bodies.** Madin Darby canine kidney cells and urothelial cancer line T24 cells were grown in Eagle's Minimum Essential Medium RPMI medium respectively, with final 10% v/v fetal bovine serum, 0.1 mM non-essential amino acids, 2 μM L-glutamine and 100 U/mL penicillin/Streptomycin in T75 flasks (Thermo Fischer Scientific, US). For induction of apoptosis, cells were incubated with 1 umol/L staurosporine (S5921, Sigma Aldrich) for 12h. Cells were harvested by trypsinisation and centrifuged for 15 minutes at 1550 g. The supernatant of initial centrifugation containing apoptotic bodies and microvesicles was divided in 1 ml volumes and pipetted into 1.8 ml centrifugation tubes and centrifuged for 1 h at 25,000 and 20˚C. Sediments were collected and washed with repeated centrifugation in isoosmolar phosphate buffered saline. Final suspension of 250 ml of gathered microvesicles was centrifuged for 30 min at 20˚C and 17,570 g. Pellets were resuspended and fixed in 4% buffer solution of formaldehyde for 1 h at room temperature, and centrifuged with corresponding speeds (ie. 1550 and 17570 g). Pelets were saved in fixative at 4˚C until use.

**Isolation of cellular micro and nano particles from blood.** The study involving human blood samples was reviewed and approved by an institutional review board Slovenian National Medical Ethical Committee, number 82/07/14 before the study began. Blood was collected by a 21-gauge needle (length 70 mm, inner radius 0.4 mm, Microlance, Becton Dickinson, NJ) in 2.7 mL evacuated tubes (BD Vacutainers, Becton Dickinson, CA) containing 270 μL trisodium citrate. Centrifugation of the samples started within 20 minutes after acquisition of the sample. To separate erythrocytes from plasma, the samples were centrifuged at 1550 × g for 20 minutes in a Centric 400/R centrifuge (Domel, Železniki, Slovenia). The upper 250 μL of plasma was slowly removed from each tube by using a tip with a wide opening and placed in a 1.5 mL Eppendorf tube. The samples were then centrifuged at 17570 × g for 30 minutes in a Centric 200/R centrifuge (Domel, Železniki, Slovenia). The supernatant (225 μL) was discarded and the pellet (25 μL) resuspended in 225 μL citrated phosphate-buffered saline. The samples were centrifuged again at 17570×g for 30 minutes, the supernatant (225 μL) discarded leaving about 30 μL of the pelet from each epruvette. Pelets were saved in fixative at 4˚C until use.

**Preparation of giant phospholipid vesicles.** The vesicles were prepared at room temperature by electroformation. In the procedure, 20 μl of phospholipid 1-palmitoyl-2-oleoyl-sn-glycero-3-phosphocholine (Avanti Polar Lipids) dissolved in 2:1 chloroform/methanol, was spread over a pair of platinum electrodes. The solvent was allowed to evaporatefor two hours. The electrodes were placed into the electroformation chamber which was then filled with 2 ml of 0.2 M sucrose solution. An alternating electric field (1 V/mm, frequency 10 Hz) was applied for 2 h.Then, the a.c. field was reduced to 0.75 V/mm, 5Hz, applied for 15 min, to 0.5 V/mm, 2 Hz, applied for 15 min, and to 0.25 V/mm, 1 Hz, applied for 30 min. After the electroformation, the content of the chamber was poured out into a plastic beaker. Then, the chamber was rinsed with 2 ml of 0.2 M solution of sugar of different molecular weight (glucose) and the

contents of the chamber were added to the solution in the plastic beaker and gently mixed by turning the beaker upside down. The solution containing the vesicles was immediately after the formation placed into the observation chamber made by a pair of cover glasses and sealed by silicon grease. The vesicles were observed by the inverted microscope Zeiss IM 35 with the phase contrast optics.

**Preparation of samples for scanning electron microscopy.** The samples were incubated in 0.1% glutaraldehyde at room temperature and centrifuged at 1550 g and 37˚C for 10 minutes. The supernatant was exchanged with phosphate and citrated phosphate buffered saline, samples were vortexed, centrifuged at 1550 g and 37˚C for 10 minutes and fixed in 2% glutaraldehyde for an hour. Fixed samples were washed by exchanging supernatant with citrated phosphate—buffered saline and incubated for 20 minutes at room temperature. This procedure was repeated 4 times while the last incubation was performed overnight at 8˚C. To dry the samples, citrated phosphate-buffered saline was exchanged gradualy by acetone. To avoid osmotic shock, acetone concentration was upgraded (50%, 60%, and 90%). The final step was performed in 100% acetone for 1 h with two exchanges of acetone. Samples were dried with liquid $CO_2$ at critical point. Dried samples were sputtered with gold to be observed by a LEO Gemini 1530 (LEO, Oberkochen, Germany) scanning electron microscope.

**Models.** The equilibrium shape of the vesicle corresponds to the minimum of its membrane free energy (Eq (3)). For uni–component membrane, the mixing entropy term in the expression of the free energy of the membrane (2.nd term in the right side of the Eq (3)) is absent while there is only one contribution to the sum of the first term. Further, it was assumed that for large anisotropy, the argument of the hyperbolic cosinus is large, therefore ln (cosh ($d_{eff}$/2)) is approximated by $d_{eff}$/2 while free energy follows from Eqs (1)–(3),

$$F = k_H \int (H - H_m)^2 \, dA + k_D \int (D - D_m)^2 \, dA \tag{6}$$

where $k_H = \xi/2$ and $k_D = (\xi + \xi^*)/2$. Also it was taken that $k_H = k_D$.

For bi–component membrane the respective expression is

$$F = k_{1,H} \int m_1 (H - H_{1,m})^2 \, dA + k_{1,D} \int m_1 (D - D_{1,m})^2 \, dA + k_{2,H} \int m_2 (H - H_{2,m})^2 \, dA$$
$$+ k_{2,D} \int m_2 (D - D_{2,m})^2 \, dA + kT \int \left( m_1 \ln \left( \frac{m_1}{m} \right) + m_2 \ln \left( \frac{m_2}{m} \right) \right) \, dA \tag{7}$$

with $k_{1,H} = k_{1,D} = k_1$ and $k_{2,H} = k_{2,D} = k_2$.

In numerical simulations, we consider closed axisymmetric membranes exhibiting spherical topology. The shell surface is assumed to be a surface of revolution with rotational symmetry about the $z$-axis within the Cartesian coordinate system $(x, y, z)$ which is defined by the unit vectors $(\mathbf{e}_x, \mathbf{e}_y, \mathbf{e}_z)$. Membrane surfaces are constructed by the rotation of the profile curve about the $\mathbf{e}_z$ axis by an angle $u = 2\pi$ (see Fig 9).

The position vector $\boldsymbol{r}$ of a generic point lying on an axisymmetric membrane surface can thus be written as [71]:

$$\boldsymbol{r} = \rho(s)\cos(u)\mathbf{e}_x + \rho(s)\sin(u)\mathbf{e}_y + z(s)\mathbf{e}_z \tag{8}$$

where $\rho(s)$ and $z(s)$ are variational parameters defining the profile curve in the $(x, z)$—plane and $s$ stands for the arc length of the profile curve (Fig 9). On surfaces of revolution, parallels and meridians are lines of principal curvature. We set that the principal directions $(\mathbf{e}_1, \mathbf{e}_2)$ point along meridians $u = const$ while for parallels, $s = const$. In this formulation, the local

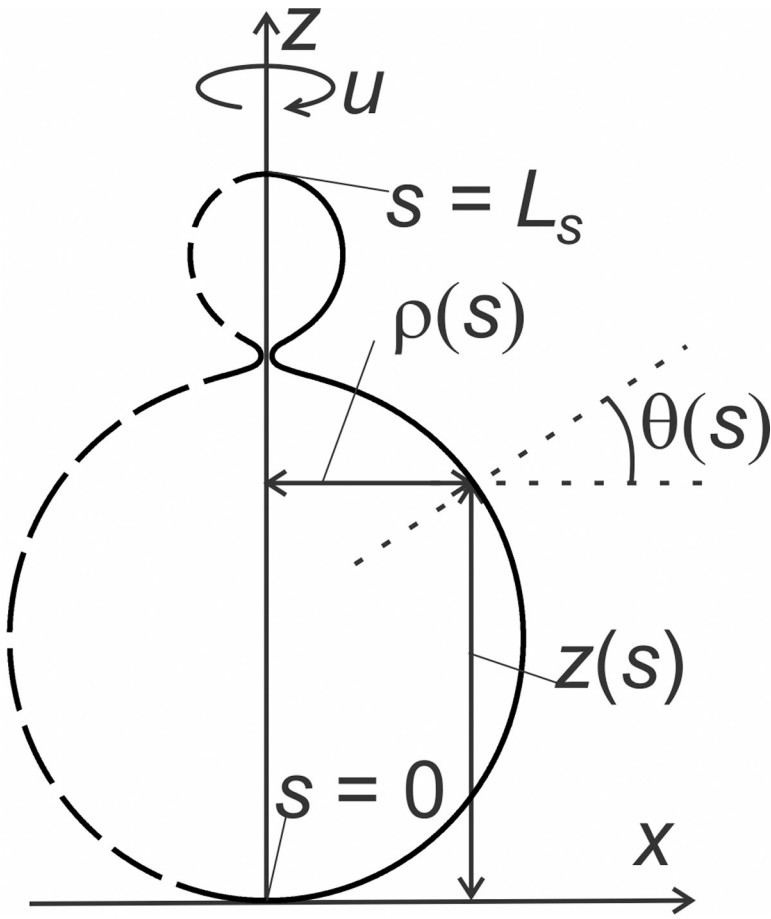

**Fig 9. Parametrization of the shape.** Profile curve in the $(x, z)$—plane, where $z(s)$ represents the height and $\rho(s)$ the radius of the shape profile at the given value of the arc length $s$. $L_s$ stands for the total length of the shape profile, $u$ is the angle of rotation about the $z$—axis and $\theta(s)$ represents the angle formed by the tangent to the profile curve and the plane that is perpendicular to the axis of rotation $\mathbf{e}_z$.

principal curvatures of the surface were calculated as [71]

$$C_1 = \frac{\mathrm{d}\theta(s)}{\mathrm{d}s}, \; C_2 = \frac{\sin\theta(s)}{\rho(s)} \tag{9}$$

where $\theta(s)$ stands for the angle of the tangent to the profile curve with the plane that is perpendicular to the axis of rotation $\mathbf{e}_z$ (Fig 9).

The profile curve of axisymmetric surface was calculated by [71–74],

$$\rho(s) = \int_0^s \cos\theta(s')\mathrm{d}s', \; z(s) = \int_0^s \sin\theta(s')\mathrm{d}s'. \tag{10}$$

The integration was performed from 0 to $s$. The position vector $\mathbf{r}$ of a generic point on axisymmetric surface was obtained by inserting Eq (10) into Eq (8). The boundary conditions for closed and smooth surfaces were as follows: $\theta(0) = 0$, $\theta(L_s) = \pi$, $\rho(0) = \rho(L_s) = 0$, where $L_s$ represents the length of the profile curve [71–74]. The function describing the angle $\theta(s)$ was

approximated by the Fourier series [71–74]

$$\theta(s) = \theta_0 \frac{s}{L_s} + \sum_{i=1}^{N} a_i \sin\left(\frac{\pi}{L_s} i \cdot s\right) \quad (11)$$

where $N$ is the number of Fourier modes, $a_i$ are the Fourier amplitudes, and $\theta_0 = \theta(L_s) = \pi$ is the angle at the north pole of the axisymmetric membrane surface. The membrane free energy $F$ (see Eqs (6) or (7)) is therefore a function of the Fourier amplitudes $a_i$ and the shape profile length $L_s$. Equilibrium closed membrane shapes were calculated by the numerical minimization of the function of many variables [71–74]. During the minimization procedure, the membrane surface area $A$ and the volume $V$ were kept constant in order to set a fixed value of the membrane reduced volume $v$. Furthermore, the dimensionless average mean curvature $< h >$ was also kept constant.

## Acknowledgments

Authors are indebted to Anna Romolo for technical assistance in preparing the manuscript.

## Author Contributions

**Conceptualization:** Veronika Kralj-Iglič, Gabriella Pocsfalvi, Henry Hägerstrand, Aleš Iglič.

**Data curation:** Luka Mesarec, Vid Šuštar, Henry Hägerstrand.

**Formal analysis:** Veronika Kralj-Iglič, Gabriella Pocsfalvi, Luka Mesarec, Vid Šuštar, Henry Hägerstrand, Aleš Iglič.

**Funding acquisition:** Veronika Kralj-Iglič, Gabriella Pocsfalvi, Henry Hägerstrand, Aleš Iglič.

**Investigation:** Veronika Kralj-Iglič, Gabriella Pocsfalvi, Luka Mesarec, Vid Šuštar, Henry Hägerstrand, Aleš Iglič.

**Methodology:** Veronika Kralj-Iglič, Luka Mesarec, Vid Šuštar, Henry Hägerstrand, Aleš Iglič.

**Project administration:** Veronika Kralj-Iglič, Gabriella Pocsfalvi, Aleš Iglič.

**Resources:** Veronika Kralj-Iglič, Henry Hägerstrand, Aleš Iglič.

**Software:** Luka Mesarec.

**Supervision:** Veronika Kralj-Iglič, Gabriella Pocsfalvi, Henry Hägerstrand, Aleš Iglič.

**Validation:** Veronika Kralj-Iglič, Gabriella Pocsfalvi, Vid Šuštar, Aleš Iglič.

**Visualization:** Veronika Kralj-Iglič, Gabriella Pocsfalvi, Luka Mesarec, Vid Šuštar, Henry Hägerstrand.

**Writing – original draft:** Veronika Kralj-Iglič, Gabriella Pocsfalvi.

**Writing – review & editing:** Veronika Kralj-Iglič, Gabriella Pocsfalvi, Luka Mesarec, Vid Šuštar, Henry Hägerstrand, Aleš Iglič.

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
