## [Decision Letter · Decision Letter 0]

7 Dec 2020

PONE-D-20-30737

Minimizing isotropic and deviatoric membrane energy – an unifying formation mechanism of different cellular membrane nanovesicle types

PLOS ONE

Dear Dr. Kralj-Iglič,

Thank you for submitting your manuscript to PLOS ONE. I have completed the evaluation of your manuscript with the comments from the reviewer. As there are few typos in the manuscript, minor revision is required. I invite you to submit a revised version of the manuscript that addresses the points raised during the review process.

We look forward to receiving your revised manuscript.

Kind regards,

Xin Yi

Academic Editor

PLOS ONE

Journal Requirements:

Reviewers' comments:

Reviewer's Responses to Questions

**Comments to the Author**

1. Is the manuscript technically sound, and do the data support the conclusions?

Reviewer #1: Yes

2. Has the statistical analysis been performed appropriately and rigorously? 

Reviewer #1: Yes

3. Have the authors made all data underlying the findings in their manuscript fully available?

Reviewer #1: Yes

4. Is the manuscript presented in an intelligible fashion and written in standard English?

Reviewer #1: Yes

5. Review Comments to the Author

Reviewer #1: The manuscript “Minimizing isotropic and deviatoric membrane energy – an unifying formation mechanism of different cellular membrane nanovesicle types” describes the formation of the nanovesicles within the deviatoric curvature model. The deviatoric curvature model is an extension of the commonly studied spontaneous curvature model. It allows for accounting the shape and orientation of the membrane constituents. Therefore, it gives more freedom in interpretation of the experimental results. The authors try to explain the experimental observation on formation of nanovesicles based on the results of the numerical calculations with a specific set of parameters for the deviatoric curvatures model. The deviatoric model is not so commonly used as the spontaneous curvature model. Such new results may be of interest to the scientific community, and may be helpful in understanding the processes which are encountered in biological membranes. I think that the results presented in the manuscript have sufficient element of novelty to be published in PlosOne.

Minor comment: in eq (7), line 670, there is a typo. There should be (m2/m) instead of (m1/m) in the second lg() of entropy of mixing term

6. PLOS authors have the option to publish the peer review history of their article (what does this mean?). If published, this will include your full peer review and any attached files.

Reviewer #1: No

---

## [Author Response · Author response to Decision Letter 0]

15 Dec 2020

Dear prof. Yi,

thank you for informing us that our manuscript can be considered for publication after a minor revision. We thank the reviewer, especially for thorough reading of the work and pointing to the negligent mistake in one of the equations. 

Comment of the reviewer: in eq (7), line 670, there is a typo. There should be (m2/m) instead of (m1/m) in the second lg() of entropy of mixing term

Answer: We have corrected the mistake (previously in line 670 and in the new version in line 707). It is highlighted yellow in the marked version.

Also, we have found 4 typing errors in further reading of the manuscript and have corrected them. We have followed the instructions for preparation of the manuscript form and inserted figure captions at the places where the figures should appear (after the paragraph where they were first mentioned). We have revised the titles of the figures. We have formatted the headings as required. These changes have changed the numbering of the text lines with respect to the previous version. All the corrections are marked yellow. 

We hope that the manuscript is appropriate for publication. Also, we hope that this work, if published, will bring something meaningful to the scientific society.

Sincerely yours, 

Veronika Kralj-Iglič, in behalf of all authors

---

## [Editor Report · Decision Letter 1]

17 Dec 2020

Minimizing isotropic and deviatoric membrane energy – an unifying formation mechanism of different cellular membrane nanovesicle types

PONE-D-20-30737R1

Dear Dr. Veronika Kralj-Iglič,

We are pleased to inform you that your manuscript has been judged scientifically suitable for publication and will be formally accepted for publication once it meets all outstanding technical requirements.

Within one week, you will receive an e-mail detailing the required amendments. When these have been addressed, you will receive a formal acceptance letter and your manuscript will be scheduled for publication.

If your institution or institutions have a press office, please notify them about your upcoming paper to help maximize its impact. If they will be preparing press materials, please inform our press team as soon as possible -- no later than 48 hours after receiving the formal acceptance. Your manuscript will remain under strict press embargo until 2 pm Eastern Time on the date of publication. For more information, please contact onepress@plos.org.

Kind regards,

Xin Yi

Academic Editor

PLOS ONE

---

## [Editor Report · Acceptance letter]

21 Dec 2020

PONE-D-20-30737R1 

Minimizing isotropic and deviatoric membrane energy – an unifying formation mechanism of different cellular membrane nanovesicle types 

Dear Dr. Kralj-Iglič:

I'm pleased to inform you that your manuscript has been deemed suitable for publication in PLOS ONE. Congratulations! Your manuscript is now with our production department. 

Kind regards, 

on behalf of

Dr. Xin Yi 

Academic Editor

PLOS ONE